# Protocol for a realist synthesis of health systems responsiveness in low-income and middle-income countries

Tolib Mirzoev [ID],[1] Anna Cronin de Chavez [ID],[1] Ana Manzano [ID],[2]
Irene Akua Agyepong,[3] Mary Eyram Ashinyo [ID],[4] Anthony Danso-Appiah,[5]
Leveana Gyimah,[6] Lucy Yevoo,[3] Elizabeth Awini,[3] Bui Thi Thu Ha,[7]
Trang Do Thi Hanh,[8] Quynh-Chi Thai Nguyen,[9] Thi Minh Le,[7] Vui Thi Le,[7]
Joseph Paul Hicks [ID],[10] Judy M Wright [ID],[11] Sumit Kane [ID][12]

For numbered affiliations see end of article.

**Correspondence to**
Professor Tolib Mirzoev;
tolib.mirzoev@lshtm.ac.uk

## ABSTRACT

**Introduction** Health systems responsiveness is a key objective of any health system, yet it is the least studied of all objectives particularly in low-income and middle-income countries. Research on health systems responsiveness highlights its multiple elements, for example, dignity and confidentiality. Little is known, however, about underlying theories of health systems responsiveness, and the mechanisms through which responsiveness works. This realist synthesis contributes to bridging these two knowledge gaps.

**Methods and analysis** In this realist synthesis, we will use a four-step process, comprising: mapping of theoretical bases, formulation of programme theories, theory refinement and testing of programme theories using literature and empirical data from Ghana and Vietnam. We will include theoretical and conceptual pieces, reviews, empirical studies and grey literature, alongside the primary data. We will explore responsiveness as entailing external and internal interactions within health systems. The search strategy will be purposive and iterative, with continuous screening and refinement of theories. Data extraction will be combined with quality appraisal, using appropriate tools. Each fragment of evidence will be appraised as it is being extracted, for its relevance to the emerging programme theories and methodological rigour. The extracted data pertaining to contexts, mechanisms and outcomes will be synthesised to identify patterns and contradictions. Results will be reported using narrative explanations, following established guidance on realist syntheses.

**Ethics and dissemination** Ethics approvals for the wider RESPONSE (Improving health systems responsiveness to neglected health needs of vulnerable groups in Ghana and Vietnam) study, of which this review is one part, were obtained from the ethics committees of the following institutions: London School of Hygiene and Tropical Medicine (ref: 22981), University of Leeds, School of Medicine (ref: MREC19-051), Ghana Health Service (ref: GHS-ERC 012/03/20) and Hanoi University of Public Health (ref: 020-149/DD-YTCC).

We will disseminate results through academic papers, conference presentations and stakeholder workshops in Ghana and Vietnam.

## Strengths and limitations of this study

► The review will identify, and bridge, the knowledge gaps in the peer-reviewed and grey literature on (a) theoretical underpinnings of health systems responsiveness and (b) the mechanisms through which responsiveness works within the contexts of low-income and middle-income countries.

► The iterative nature of searches, screening, data extraction and quality assessments is a key strength of our approach, which allows deep engagements with the literature across multiple disciplines and settings.

► However, a potentially large scope of literature on various elements of health systems responsiveness would require careful management of the research team workload, through prioritisation of programme theories in the searches and the analysis.

► The composition of a large multicountry team will enable drawing on different context-specific understandings of health systems responsiveness.

► However, the large cross-country review team will require careful coordination, to ensure representation of different perspectives in the analysis and shared intellectual leadership and conclusions.

**PROSPERO registration number** CRD42020200353.
Full record: https://www.crd.york.ac.uk/prospero/display_record.php?ID=CRD42020200353.

## INTRODUCTION

Health systems responsiveness is a key goal of any national health system and is "…*when institutions… are cognisant and respond appropriately to the universally legitimate expectations of individuals… safeguarding of rights of patients to adequate… care*"[1](p3) People, especially the most vulnerable, are more likely to use services if health systems are responsive to their expectations.[2–4] For example, evidence shows that people's trust in their health systems and feeling secure within health

facilities can improve the uptake of maternal health services among pregnant women in Nigeria.[5 6] Similarly, issues of trust determine utilisation of maternal healthcare by women from ethnic minorities in remote areas of Vietnam.[7 8] Thus, responsive health systems can facilitate improvements in uptake of healthcare services and ensure adherence to treatment and ultimately contribute towards enhanced patient welfare and equitable improvements in population health.[3 9]

When considering all health systems goals (which include improved health, fair financial contribution and efficiency),[10 11] responsiveness is the least studied, particularly in low and middle-income countries (LMICs).[12 13] Research on measuring health systems responsiveness used a survey toolkit from the WHO, which highlights seven elements of responsiveness: dignity, autonomy, confidentiality, prompt attention, quality of amenities, access to support networks and choice of service provider.[1 2 4 14–17] These elements stem from literature on quality of care and patient satisfaction.[18] The literature also highlights that interpretations of responsiveness are *context-sensitive* (eg, expectations of dignity reflect political, democratic and policy environment)[2] *vary across actors* (eg, patients and providers, reflecting their different powers)[3 19] and *health facilities* (eg, public/private).[2 3] Responsiveness, therefore, is arguably a socially constructed, rather than an 'absolute' and 'universally normative' concept. Recent work also emphasises that interactions between people and their health systems are central to understanding health systems responsiveness[12] and improving responsiveness should, therefore, address both the 'people' and the 'systems' sides of such interactions.[12 20 21]

Little is known, however, about: (a) theoretical underpinnings of the current interpretation of health systems responsiveness and (b) underlying mechanisms through which health systems responsiveness works for different actors (such as communities, health workers, managers) and under which conditions. This realist synthesis (RS) will contribute to bridging these knowledge gaps through advancing theorisation of health system responsiveness and providing an in-depth understanding of key mechanisms of how responsiveness works for different health systems actors. We hope that the results of this review will ultimately inform improvements to health systems responsiveness in Ghana, Vietnam and other LMICs.

This RS is being undertaken as part of the wider RESPONSE study, which is a mixed-methods realist evaluation of health systems responsiveness in Ghana and Vietnam. RESPONSE seeks to contribute to improving health systems responsiveness in LMICs through case studies of addressing neglected health needs of vulnerable groups in Ghana and Vietnam. As described in the RESPONSE protocol available elsewhere,[22] in Ghana, the study will be implemented in Greater Accra Region and in Vietnam, we will work in Bắc Giang Province. The study is being implemented jointly by the London School of Hygiene and Tropical Medicine, University of Ghana, Ghana Health Service, Mental Health Authority of Ghana, Hanoi University of Public Health, University of Leeds and University of Melbourne.

## Aim and questions

This RS will deepen the understanding of theoretical foundations of health systems responsiveness and the mechanisms through which health systems responsiveness works (or not) for different health systems actors in different contexts. Through developing, testing and validating programme theories, we will identify and explain which contexts trigger specific mechanisms through which health systems responsiveness can produce the intended or unintended outcomes. In the process, we will also highlight any gaps in the existing literature following our review. This RS will answer the following questions:

1. Which substantive theories underpin the understanding of health systems responsiveness in the literature, and how do they inform these interpretations?
2. In what way does health systems responsiveness work for different health systems actors (service users, providers and managers) in the contexts of LMIC?

The main result from this review will be detailed narrative explanations[23] of how health systems responsiveness works for different health systems actors in LMICs. These explanations will draw on available theoretical and empirical literature and insights into fieldwork in Ghana and Vietnam.

## METHODS AND ANALYSIS

This RS uses a realist approach, which helps understand complex programmes by identifying how the multiple components interact in nonlinear ways.[24 25] It is guided by an overall question of 'what works for whom, under which circumstances and why'[25] and is particularly appropriate for exploring the socially constructed phenomena. A realist approach recognises micro (individual), meso (organisational) and macro (systemic) contexts (Cs) in triggering the mechanisms, which comprise reasoning and resources (Ms), to produce intended or unintended outcomes (Os)—altogether known as CMO (Context-Mechanism-Outcome) configurations.[26–31] Such an approach is particularly suited to understanding the complexity of health systems responsiveness in diverse settings.

RS, which applies realist logic to a systematic review methodology, provides in-depth understanding of complex phenomena through articulating theories shaped as CMO configurations that explain why, when and for whom the programmes or interventions work.[23 24] However, in contrast with traditional systematic reviews, realist reviews are "…*not a method or formula, but a logic of enquiry that is inherently pluralist and flexible, embracing both qualitative and quantitative, formative and summative, prospective and retrospective…*"p32. Furthermore, RSs are more inclusive of the types of studies that can be included[23] and often incorporate more substantial consideration of grey literature.[32]

Conceptually, RS involves broad phases that mirror the realist evaluation process: formulating questions, developing initial theories, testing and refining theories,[33] all culminating in the final narrative.[23] Specific steps in conducting RS are close to those of the systematic reviews: identifying review questions, searching for primary studies, quality assessment, data extraction, synthesising results and dissemination.[23 24] However, these steps are not linear and, in contrast with traditional systematic reviews, overlap and are highly iterative.

Another key distinctive feature of the RS approach is that further to engagements with the literature, researchers also engage with key stakeholders, for example, in formulating initial theories and developing policy recommendations.[32] However, there is no clear guidance on involvement of stakeholders in realist syntheses[34] with the RAMESES (Realist And MEta-narrative Evidence Syntheses: Evolving Standards) standards for realist syntheses being intentionally flexible and just referring to achievement of end-user relevance.[35] As a result, there is a wide variation in the degrees to which stakeholder engagements are approached in RSs, which include informal consultations in formulating initial theories and developing policy recommendations[32] and formal data collection and analysis, for example, using in-depth interviews and surveys, to inform theory refinement[36] and even theory testing.[33]

In this RS, we are guided by the RAMESES publications standards for realist syntheses[35] (see online supplemental file) and will follow a four-step process similar to Cooper *et al*,[33] which will be embedded throughout the RESPONSE study (figure 1).

*Step* 1 will involve mapping of theoretical underpinnings of health systems responsiveness in LMIC,[23 33] through initial screening of literature. During this step, we will identify and analyse exclusively theoretical literature, which can help understand, and explain, health systems responsiveness. The main output from this step will be advanced theorisation of health systems responsiveness.

During *step 2,* we will formulate initial theories of how health systems responsiveness works. We will draw on theorisation of responsiveness from the previous step, team discussions and initial consultations with key health systems actors[32 33] in Ghana and Vietnam. During this step, we will potentially widen the scope of the literature review to include empirical studies on health systems responsiveness and will also review policy documentation in Ghana and Vietnam. Initial stakeholder consultations will be in the form of in-depth interviews with purposefully identified key health systems actors (service users, communities, service providers and managers). These interviews are planned during the baseline data collection within RESPONSE to understand what responsiveness means to different health systems actors including its importance, underlying principles, components, mechanisms and intended outcomes,[22] and analysis of interview data will contribute to formulation of initial programme theories. The main output from this step will be a (longer) list of programme theories of how health systems responsiveness works in LMICs.

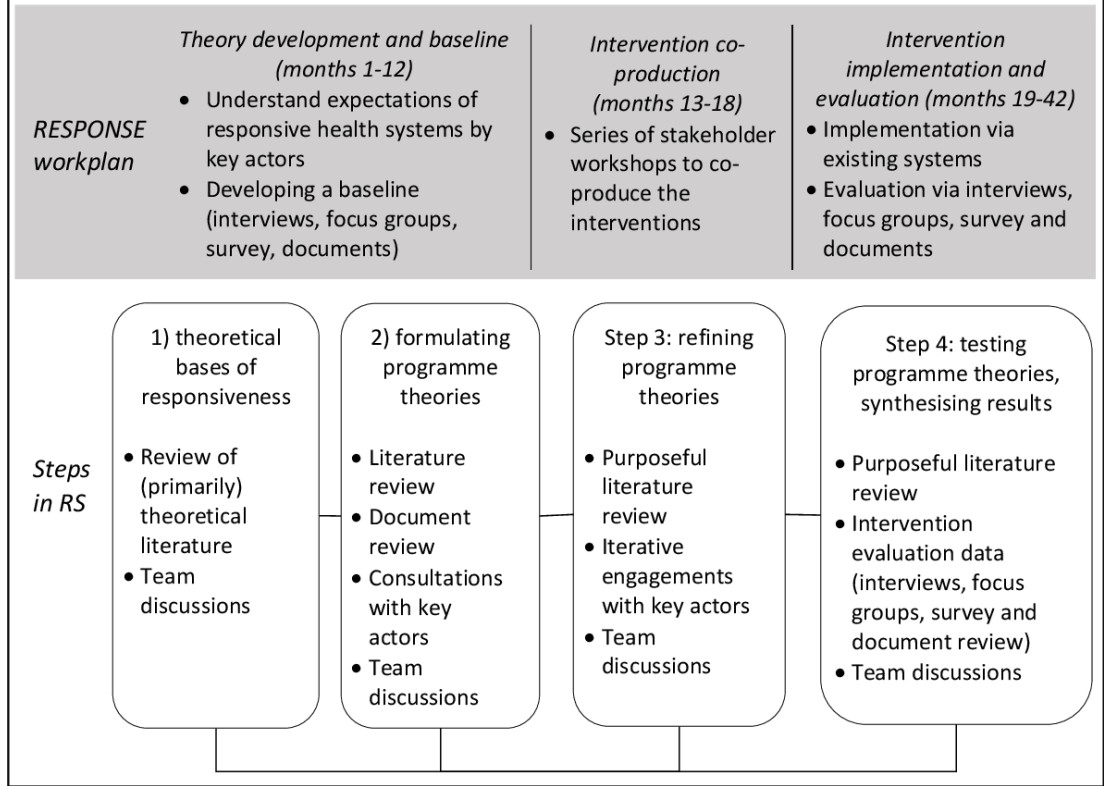

**Figure 1** Realist synthesis (RS) within the RESPONSE study.

In *step 3*, we will refine programme theories through continuing the purposeful screening of the literature alongside iterative engagements with key actors[32] in Ghana and Vietnam. Our engagements with key actors will use the planned intervention coproduction workshops. In each country, we will aim to organise four to six intervention coproduction workshops involving key actors (communities, service providers, facility managers, regional/province and national-level actors). These workshops will be led by relevant health authorities and facilitated by researchers. We will carefully document stakeholder views of causal pathways of how the interventions are intended to work to inform theory refinement, and through participant observations will also reflect on the coproduction processes.[22] The interventions to improve health system responsiveness will not be targeting individual elements of responsiveness in the WHO framework[1] but will seek to improve internal interactions (ie, within health system) and external interaction (ie, people systems), through series of participatory workshops with health workers and communities.[12 22] The main output from this step will be a (shorter) list of selected programme theories for subsequent testing.

Finally, *step 4* will entail testing of programme theories. This will be done through using insights into empirical evaluations of implemented interventions in Ghana and Vietnam and through analysing empirical literature.[33] During this step, we will review literature that supports or refutes the programme theories and will use analytical insights into empirical evaluations of the interventions. We will draw on an approach used by Cooper *et al* where interview data were integrated with evidence from the literature in testing programme theories.[33] We already plan to evaluate effectiveness of interventions within the RESPONSE,[22] and we will use this primary data set for testing our programme theories. In each country, data collection will include combinations of about 60 in-depth interviews and 4–6 focus groups with purposefully identified key health systems actors (communities and service users, service providers), a multistage clustered household community survey with sample size estimated at 562 and review of relevant policy and facility documentation. Data collection tools, such as question guides for in-depth interviews and focus groups and survey questionnaire, will be structured by the selected programme theories and the empirical data will be analysed retroductively (ie, both inductively and deductively[13 22 37] as described in the data analysis later) alongside the insights into the literature, to test programme theories. The main output from this step will comprise results synthesised into narrative explanations[23] of how health systems responsiveness works for different health systems actors in LMICs.

## Eligibility

In line with established principles on eligibility of evidence for RSs,[23 32] we will not impose any strict restrictions on the study types and will include all available studies that focus on responsiveness of health and other public systems within LMICs. Specifically, we will include: (a) theoretical and conceptual pieces, opinions and analyses, (b) reviews: systematic reviews, scoping reviews, meta-syntheses, realist syntheses, (c) empirical studies: qualitative studies, policy studies, randomised controlled trials, quasi-experimental studies and cross-sectional and cohort studies and (d) grey literature, including discussion pieces, reports, policies, plans and guidelines in Ghana and Vietnam and relevant local-level documentation within health facilities.

However, we will exclude: (a) empirical (but not theoretical) studies published before 2000, that is, the year when the WHO introduced health systems responsiveness as an explicit health systems objective, (b) studies with no full texts available for subsequent analysis and (c) studies in languages where we are unable to source translation. The review is being conducted by a team comprising speakers of English, Spanish, French, Russian and Vietnamese. While initial screening will be conducted in either of these languages, analysis will be conducted in English and we will aim to either obtain an English copy or arrange for a translation where possible.

Next, we summarise eligibility following a participants–interventions–comparison/control–outcomes approach.

Health system responsiveness affects a range of different populations such as communities, individuals, service providers, facility managers, regional or provincial and national-level policymakers.[1 12 18] Therefore, there will be no restriction on population groups. Evidence suggests that vulnerable individuals are especially dependant on the degree of health system responsiveness to their needs.[38 39] In this review, while we will not exclude any populations, we will pay particular attention to responsiveness to pregnant women suffering from mental health conditions.[22] We anticipate that this group will provide a depth of insight into responsiveness as mental health is typically the most neglected aspect of maternal health[40 41] and that responsive health systems should recognise such intersectional nature of vulnerability.

Our overall 'intervention' for improved health system responsiveness will be two types of interactions, which underpin the RESPONSE's theoretical framework[22]: internal interactions (ie, within health systems such as between service providers and managers) and external interactions (ie, between service users and health systems, typically at the point of service provision).[12]

Health system responsiveness is a complex intervention; therefore, the control is not applicable to this review. However, we will compare the CMO configurations across the different LMIC settings where it is feasible.

Our overarching outcome is improved health systems responsiveness, reflected in improved interactions, which follow the socially constructed conceptualisation of responsiveness within RESPONSE.[12 20–22]

Instead of separately considering each of seven elements of health systems responsiveness from the WHO framework[1] as distinct outcomes, we will seek to understand improvement in responsiveness of health systems

in LMICs as a whole. In the process, some of the WHO elements of responsiveness may be aggregated into fewer and broader categories. As the review progresses, further outcomes may also emerge within specific CMO configurations.

## Literature searches

Searching for theoretical and empirical evidence for realist reviews is highly iterative. Multiple searches will be conducted using a variety of appropriate search techniques.[42] Conventional database searches will be complemented with CLUSTER search techniques such as citation snowballing and project searching (eg, WHO responsiveness tool project).[43] These techniques will be used to identify related relevant studies and 'contextually rich' evidence.

Conceptually, RS involves four different types of searches at different stages of the review: a background search to get a feel of the literature, progressive searches to identify programme theories, an evidence search for evaluation studies to test chosen theories and a final search once the review is almost complete to refine theories.[32 42] We will follow such an approach throughout our four steps, our search strategy will be purposive rather than exhaustive and will evolve as our programme theories develop.

In answering the first review question (underlying theories of responsiveness), our searches will not be limited to any geographic area, whereas in answering the second review questions (how responsiveness works) will focus on the literature from LMICs. We will search combinations of:

1. Published literature from eight scientific databases: Applied Social Science Index Abstracts, CINAHL, Global Health, Maternity and Infant Care Database, Medline, PsycINFO, Scopus, Web of Science.
2. Grey literature from main gateways (eg, Eldis) and relevant organisations (eg, WHO repository, MEASURE Evaluation).
3. Health systems policy and regulatory documentation from Ghana and Vietnam.

At step 1, we will search Medline (1946+) and Google Scholar for reports containing substantive theories and frameworks. MeSHs (Medical Subject Headings), free-text words and synonyms will be used for the following search concepts (see online supplemental file for a sample Medline search strategy):

▶ *health systems responsiveness*: responsiveness/satisfaction/confidence/trust and WHO elements of responsiveness: dignity; autonomy; confidentiality; prompt attention; quality of amenities; access to support networks and choice of service provider.
▶ *Theories*: theory; framework; conceptualisation.

During step 2, we will widen the search from the previous step to also include empirical literature on health systems responsiveness within LMICs and systems policy and regulatory documentation from Ghana and Vietnam.

At steps 3 and 4, targeted searches for empirical and theoretical literature on how health systems

responsiveness works in LMICs will be determined by the selected programme theories. MeSHs, free-text words and synonyms will be used for the following search concepts:

▶ *different health systems actors*: communities/pregnant women/service providers/doctors/nurses/ health workers/ health managers/health providers/ policymakers/decision-makers/implementers/ researchers/.
▶ *LMIC* contexts, specifically sub-Saharan Africa and South East Asia.
▶ *Programme theory* being tested.

Search results will be managed using EndNote V.X9 software. Records will be uploaded into Rayyan software (https://rayyan.qcri.org/welcome) for initial screening as appropriate. Qualitative data analysis software NVIVO (version 10) will be used for organising and managing data, cross-referencing concepts, actors and populations. Data extraction will be done with Microsoft Word or Excel.

Screening in RSs is typically done throughout the review as the searches progress, rather than in one large exercise as in the systematic reviews.[23] In our review, we will conduct: (a) initial screening for theoretical frameworks and models of health systems responsiveness during step 1 and then (b) purposeful screening in steps 2–4 for developing, refining and testing programme theories. Screening in each step will be done collaboratively by the different team members, and we will ensure that a minimum of 20% of results are double screened for calibration and quality control.

## Data extraction

Realist syntheses have no standardised common data extraction form and data extraction is often combined with quality appraisal of studies.[23 24 33] As RS involves a highly iterative process with search terms continually refined, the process will be documented carefully for reporting.

During all steps, a data extraction table in Microsoft Word or Excel format will be used to track included papers, grey literature and policy documents, adapting the following headings to each step of the RS:

a. key paper identifiers (full citation).
b. Type of paper (published, grey literature).
c. Social science theory introduced and/or used.
d. Elements of health systems responsiveness addressed.[112]
e. Links to selected programme theory(ies) showing causality in CMO configurations, through identifying evidence on which key contexts trigger specific mechanisms to produce intended or unintended outcomes.

In step 1 of our RS, we will specifically focus on understanding theorisation of responsiveness and will use and potentially expand headings a–c. In steps 2–4, we will add headings d–e as applicable to stages of theory development, refinement and testing.

The primary data collection in step 4 will involve tailored topic guides for interviews and focus groups, which will be structured around the selected programme theories.

Transcripts from the interviews and focus groups will be coded by the CMO configurations (or their fragments) to help refine, discard or consolidate the causality within selected programme theories during the analysis.

## Quality assessment

Quality assessment, data extraction and collation of stakeholder opinions are typically combined in RS,[32] a process which we will also follow in our review.

All studies will be considered for their *relevance* to programme theories of health systems responsiveness, specifically ability to provide insights into contexts, mechanisms and outcomes and *rigour*, commensurate to respective study types.[23 32]

Quality appraisal in RS is usually done on case-by-case basis and we will adapt and use appropriate tools (eg, JBI (Joanna Briggs Institute) or CASP (Critical Appraisal Skills Programme) quality appraisal tools[44] and use applicable guidance (eg, RAMESES standards).[35 45] Assessing relevance and rigour will be undertaken collaboratively through regular team discussions. Quality assessment will be conducted on each fragment of evidence as it is being extracted, rather on whole studies.

## Data analysis and synthesis

We will undertake a retroductive approach to analysis meaning, combining both inductive and deductive reasoning logics together.[13 22 37] Such an approach is a distinctive feature of realist studies, which also draws on 'insights or hunches' to help identify the hidden causal forces behind the identified patterns such as outcomes.[37]

Analysis of subgroups in RS normally emerges as part of the theory elicitation process.[24 32] Therefore, it would be inappropriate to narrowly predetermine the subgroups. The literature on RS also distinguishes specific and iterative steps in the data synthesis, for example, organising extracted data into evidence tables, theming by individual reviewers, formulating chains of inference (CMO connections at theory and subtheory levels) from the themes, linking chains of inference and hypothesis formulation.[23]

Analysis of theorisation of health systems responsiveness in step 1 will not be restricted to any geographical, health condition or population subsets. As the review develops, from step two onwards we will add focus on empirical literature from the LMICs. We are particularly interested in maternal mental health as a possible condition subset, and key health systems actors (service users, service providers and managers) as a population subset. However, the subgroup analysis will be fully driven by the theory elicitation.

We will relate the results of our analyses within these subsets to the health systems responsiveness as a whole, reflected in internal and external interactions within health systems. Extracted data pertaining to contexts, mechanisms and outcomes will be synthesised to identify patterns and contradictions in the relationships between contexts, mechanisms and outcomes. These relationships will be documented and will inform initial theories from step 2, to be subsequently refined and tested during steps 3 and 4, respectively. Data synthesis will also lead us to identify social science theories that may further explain our programme theories. At this stage, middle-range theories will be identified and extracted.

Our synthesis will be specifically concerned with understanding the conditions under which health systems responsiveness works (or not) for different health systems actors. Data synthesis in RS takes several forms, essentially entailing a form of 'triangulation', bringing together information from different studies to explain why a pattern of outcomes may occur.[32 33] Our review would go further and would identify explanations for potentially contrasting (or similar) findings to identify the circumstances in which the intended mechanisms and intended outcomes occur (improved health systems responsiveness, reflected in improved interactions) and those in which unintended mechanisms and outcomes (eg, sense of poor interactions due to mutual distrust) occur.

We will specifically synthesise data along our conceptualisations of health systems responsiveness (responsiveness as a whole, internal and external interactions[12] and possibly relevant elements where applicable).[1] These will relate closely to our programme theories and further elements of data synthesis may be added to capture the causality and contingent nature of the CMO configurations. In reporting results, we will produce rich narrative explanations of our programme theories,[23] following established guidance on reporting realist syntheses.[23 24 35]

## Patient and public involvement

This review will use published data alongside primary data. We do not envisage any patient and public involvement during step 1 (theorisation of health systems responsiveness). However, patients and members of the public will be involved in steps 2–4 as part of the wider stakeholder engagement. In step 2, their involvement would be as sources of local knowledge in identifying priority areas for responsiveness during initial consultations. In step 3, their involvement would comprise more active participation in the iterative engagements during intervention coproduction. In step 4, patient and public involvement would be evaluative of the effects of interventions.

## ETHICS AND DISSEMINATION

Ethics approvals for the wider RESPONSE study, which also cover the primary data collection for the RS, were obtained from the London School of Hygiene and Tropical Medicine (ref: 22981), the University of Leeds School of Medicine (ref: MREC19-051), Ghana Health Service (ref GHS-ERC 012/03/20) and Hanoi University of Public Health (ref 020-149/DD-YTCC).

We will disseminate results through academic papers and stakeholder workshops in Ghana and Vietnam. The findings will also be presented at national and international scientific conferences, such as the biannual Global Symposia on Health Systems Research.

**Author affiliations**
¹Global Health and Development, London School of Hygiene & Tropical Medicine, London, UK
²School of Sociology and Social Policy, University of Leeds, Leeds, UK
³Research and Development Division, Ghana Health Service, Accra, Greater Accra, Ghana
⁴Department of Quality Assurance, Institutional Care Directorate, Ghana Health Service, Accra, Ghana
⁵School of Public Health, University of Ghana, Accra-Legon, Ghana
⁶Pantang Hospital, Mental Health Authority, Accra, Ghana
⁷Department of Population and Reproductive Health, Hanoi University of Public Health, Hanoi, Vietnam
⁸Department of Environmental Health, Hanoi University of Public Health, Hanoi, Viet Nam
⁹Department of Health Promotion, Hanoi University of Public Health, Hanoi, Vietnam
¹⁰Nuffield Centre for International Health and Development, University of Leeds, Leeds, UK
¹¹Leeds Institute of Health Sciences, University of Leeds, Leeds, UK
¹²Nossal Institute for Global Health, University of Melbourne Queen's College, Parkville, Victoria, Australia

**Contributors** TM and SK conceived the study; TM, ACdC, AM, IAA, MEA, AD-A, LG, LY, EA, BTTH, TDTH, TML, VTL, JH, JMW, SK substantially contributed to the design of the work, jointly wrote the protocol, read and approved the final version of the manuscript.

**Funding** The research protocol reported in this paper received funding from the Joint MRC/ESRC/DFID/Wellcome Health Systems Research Initiative (grant ref: MR/T023481/1).

**Competing interests** None declared.

**Patient consent for publication** Not required.

**Provenance and peer review** Not commissioned; externally peer reviewed.

**ORCID iDs**
Tolib Mirzoev http://orcid.org/0000-0003-2959-9187
Anna Cronin de Chavez http://orcid.org/0000-0002-4050-4276
Ana Manzano http://orcid.org/0000-0001-6277-3752
Mary Eyram Ashinyo http://orcid.org/0000-0002-8493-9378
Joseph Paul Hicks http://orcid.org/0000-0002-0303-6207
Judy M Wright http://orcid.org/0000-0002-5239-0173
Sumit Kane http://orcid.org/0000-0002-4858-7344

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
