## [Reviewer comments · BMJ Open]

ARTICLE DETAILS

TITLE (PROVISIONAL)	Protocol for a realist synthesis of health systems responsiveness in low- and middle-income countries
AUTHORS	Mirzoev, Tolib; Cronin de Chavez, Anna; Manzano, Ana; Agyepong, Irene; Ashinyo, Mary Efram; Danso-Appiah, Anthony; Gyimah, Leveana; Yevo, Lucy; Awini, Elizabeth; Bui, Ha Thi Thu; Do Thi Hanh, Trang; Nguyen, Quynh-Chi; Le, Thi; Thi Le, Vui; Hicks, Joseph; Wright, Judy; Kane, Sumit

VERSION 1 – REVIEW

REVIEWER	Lahariya, Chandrakant G. R. Medical College, Community Medicine
REVIEW RETURNED	20-Jan-2021

GENERAL COMMENTS	The paper need major re-writing and clear writing of hypothesis. In the current version, it is not very reader friendly and key messages are not being communicated.
--

REVIEWER	Dieleman, Marjolein Vrije University , Earth and Life Sciences
REVIEW RETURNED	29-Jan-2021

GENERAL COMMENTS	The protocol addresses an interesting and important topic and the choice of RS is appropriate. It is also interesting to read that local stakeholders in Vietnam and Ghana will be actively involved in the process. However, the structure and logic of presentation of especially the methods-sections of the manuscript could be improved. Currently, it is not very easy to understand the different steps and its details, and the text has some repetitions. In addition, in several areas further clarifications are required to better understand how the RS will be conducted: 1) Regarding the concept of health systems responsiveness: This is a very broad term, and each of the seven domains can be a topic for a review and each can have different ways of operationalization, using different theoretical frameworks: can the authors clarify how they will address the vastness of this topic and make it manageable for their review? The authors describe in the rationale that responsiveness is socially-constructed, can they relate this more explicitly to their approach to show how they apply their understanding of HSR in the RS? 2) Regarding the objective and research questions
---

	The research objectives are broad and cover LMIC, whereas the overall objective states it is specifically aiming to identify how this works for health system actors in Ghana and Vietnam. Please align. The authors are requested to clarify the population group: in the objective “ the most vulnerable pregnant women” is mentioned, but it is unclear how this group is defined (who are the most vulnerable) , and why this focus is made. It would be helpful to explain this focus in the introduction and further define when “participants” are described. 3) Approach to RS It is interesting to involve key system actors from Ghana and Vietnam to discuss the findings of step 1 and 2 of the RS. However, it is unclear how testing and refining theories in Ghana and Vietnam are part of the RS, as from the text it seems that the focus of this step is not on empirical data identified in published and grey literature that is discussed with stakeholders, but on data from the field. It is important that the authors better explain their method and provide a rationale for this. In addition, in the section on research aims testing and refining theories is described as outcome of the review, and in the methods it is described as part 4 of the RS: this is unclear. Moreover, the overall text on methods would benefit from a clearer description of realist synthesis and the CMO-concepts. 4) Embedding RS in overall study It would be helpful to visualize the different components of the RESPONSE-study and how the RS fits in here. The current text is generally unclear as to how the steps of RS fit in the overall project, but in particular the relation between step 4 in the RS and phase 3 of the overall project requires more clarity. 5) Eligibility: Please add a heading “Literature search” as the heading “Eligibility” seems to refer to what documents will be included and excluded; and explain what the authors address in this section upfront. The headings that are provided (participants, interventions, outcome) require a definition in the context of this RS. Furthermore: - It is unclear how the authors will decide if the review will contribute to testing theories in Ghana and Vietnam, please clarify. - Interventions: is this a definition of the concept or what type of interventions will be searched for in the literature or both? - Outcome: please clarify how this RS will remain manageable: the current description gives the impression of a very vast search, especially as the review will use the seven domains of WHO (which can each be defined and conceptualised in different ways) and a wide range of system actors (See also comment 1). 6) Data extraction Please further clarify how you will extract links to program theory and showing causality. More explanation on the use of community surveys and FGDs to inform CMO and how this links to the literature review, and in particular to data extraction is required. Please clarify retroductive analysis and coding: how are these related?
--	---

	7) Synthesis The sentence: Thus, it seeks to provide etc.; suggestion to leave this sentence out. 8) Patient and public involvement Interesting to have population groups involved in the process, at the same time: it is unclear which groups will be involved, or -if this can not yet be defined- what the process of selecting these groups is, and what the sampling and recruitment process will be. Lastly: the authors are requested to edit the manuscript and address some grammar errors and long sentences to further improve the reading
--	---

VERSION 1 – AUTHOR RESPONSE

Comment	Actions taken and response ¹
Reviewer: 1 Dr. Chandrakant Lahariya	
The paper need major re-writing and clear writing of hypothesis. In the current version, it is not very reader friendly and key messages are not being communicated	We take the point about the reader friendliness and have now revised the paper throughout. Due to the nature of realist syntheses, the hypotheses usually emerge from the review itself (as programme theories), rather than pre-determined. In addressing this comment, we have also revisited review aim and questions (pp. 3-4) and the key messages are now clearer in the Strengths and limitations section (pp.2-3).
Reviewer: 2 Dr. Marjolein Dieleman	
The protocol addresses an interesting and important topic and the choice of RS is appropriate. It is also interesting to read that local stakeholders in Vietnam and Ghana will be actively involved in the process.	Thank you, no action was required in response to this comment
However, the structure and logic of presentation of especially the methods-sections of the manuscript could be improved. Currently, it is not very easy to understand the different steps and its details, and the text has some repetitions.	We take this point, and have now articulated the different steps and their details more clearly (pp. 5-6, and also Figure 1) and removed repetitions with cross-references throughout
In addition, in several areas further clarifications are required to better understand how the RS will be conducted: 1) Regarding the concept of health systems responsiveness:	We take this point, and our approach to making this review manageable is to be driven by the selected programme theories from the review. In other words, we will not be conducting seven reviews covering each domain (we aligned terminology to 'elements'

¹ Page numbers refer to a clean unmarked copy or 'simple markup' view in Word

Comment	Actions taken and response ¹
This is a very broad term, and each of the seven domains can be a topic for a review and each can have different ways of operationalization, using different theoretical frameworks: can the authors clarify how they will address the vastness of this topic and make it manageable for their review?	throughout) separately, but on the systems responsiveness as a whole, reflected in internal and external interactions from our conceptual framework. As a result, we may end up with fewer but broader elements, or indeed may replace or add more elements. This is now clarified in the description of the four steps and outcomes (pp. 6-8) and Data analysis (pp.10-11).
The authors describe in the rationale that responsiveness is socially-constructed, can they relate this more explicitly to their approach to show how they apply their understanding of HSR in the RS?	As recommended, we have now linked more clearly the socially-constructed nature of responsiveness with a realist nature of our review (Methods and analysis, p. 5).
2) Regarding the objective and research questions The research objectives are broad and cover LMIC, whereas the overall objective states it is specifically aiming to identify how this works for health system actors in Ghana and Vietnam. Please align.	This review is being conducted as part of the wider RESPONSE study, so while the literature part of the review will focus on LMICs, the stakeholder engagements and primary data will be specifically from the two countries. We have clarified this in explaining the four steps of the review (pp. 6-7).
The authors are requested to clarify the population group: in the objective “ the most vulnerable pregnant women” is mentioned, but it is unclear how this group is defined (who are the most vulnerable) , and why this focus is made. It would be helpful to explain this focus in the introduction and further define when “participants” are described.	The population group for the study (including this review) is explained in the published protocol for RESPONSE study, and in addressing this comment we have included more details on the population (on p.7).
3) Approach to RS It is interesting to involve key system actors from Ghana and Vietnam to discuss the findings of step 1 and 2 of the RS. However, it is unclear how testing and refining theories in Ghana and Vietnam are part of the RS, as from the text it seems that the focus of this step is not on empirical data identified in published and grey literature that is discussed with stakeholders, but on data from the field. It is important that the authors better explain their method and provide a rationale for this. In addition, in the section on research aims testing and refining theories is described as outcome of the review, and in the methods it is described as part 4 of the RS: this is unclear.	The review will be conducted alongside the other components of the RESPONSE study, so theory testing will utilise both literature and empirical field data. This is in line with similar realist syntheses, which also combined empirical data from the literature and primary data from the field - explained and referenced in the second paragraph of the Methods and analysis (p.5) and description of the step 4 on p. 7. In addressing this comment, we have also modified the wording of the Aims (pp. 4-5) which makes it clear that theory testing and refinement are part of the review.
Moreover, the overall text on methods would benefit from a clearer description of realist synthesis and the CMO-concepts.	As recommended, we have described the concepts of RS and CMOs more clearly in the Methods and analysis (p.5)
4) Embedding RS in overall study It would be helpful to visualize the different components of the RESPONSE-study and how the	As suggested, we have included Figure 1 (signposted on p.5 and uploaded in a separate file) to visualise these relationships, also cross-referring to the

Comment	Actions taken and response ¹
RS fits in here. The current text is generally unclear as to how the steps of RS fit in the overall project, but in particular the relation between step 4 in the RS and phase 3 of the overall project requires more clarity.	published protocol for RESPONSE study . In addressing this comment, we have also improved descriptions of how the four review steps correspond to the project timeline (p.6-7) and have removed unnecessary references to different project phases throughout.
5) Eligibility: Please add a heading “Literature search” as the heading “Eligibility” seems to refer to what documents will be included and excluded; and explain what the authors address in this section upfront. The headings that are provided (participants, interventions, outcome) require a definition in the context of this RS.	As suggested, we have added the heading ‘Literature searches’ on p.8 and also modified the remaining headings accordingly. In the process, we have also removed separate smaller subheadings – such as the PICOs, study records and screening, and restructured the material in a more reader-friendly narrative throughout pp. 7-10.
Furthermore: - It is unclear how the authors will decide if the review will contribute to testing theories in Ghana and Vietnam, please clarify.	As recommended, in explaining the four steps (pp. 6-7) we have clarified that testing of theories will inform, and will also draw upon, the intervention co-production, implementation and evaluation
- Interventions: is this a definition of the concept or what type of interventions will be searched for in the literature or both?	We will search for the underlying theories of responsiveness and how responsiveness works as per our review questions. In addressing this comment, we have clarified this in the Eligibility and Literature Searches (pp. 7-9)
- Outcome: please clarify how this RS will remain manageable: the current description gives the impression of a very vast search, especially as the review will use the seven domains of WHO (which can each be defined and conceptualised in different ways) and a wide range of system actors (See also comment 1)	As we have also explained in our response to an earlier comment 1, we will focus on responsiveness as a whole (reflected in internal and external interactions as per our theoretical framework) and not individual elements of the WHO framework. This is now clarified in the Eligibility on p.7 and Data synthesis and analysis on pp. 10-11.
6) Data extraction Please further clarify how you will extract links to program theory and showing causality. More explanation on the use of community surveys and FGDs to inform CMO and how this links to the literature review, and in particular to data extraction is required. Please clarify retroductive analyse and coding: how are these related?	As recommended, we have now explained what retroductive approach to data analysis entails in the Data Analysis and synthesis (p. 10). We have also clarified the links to the programme theories and causality in the data extraction, and that the data extraction refers to the data from the literature only, but the transcripts from FGDs and interviews will be coded by the CMO configurations.
7) Synthesis The sentence: Thus, it seeks to provide etc.,: suggestion to leave this sentence out.	As suggested, this sentence is now removed.
8) Patient and public involvement Interesting to have population groups involved in the process, at the same time: it is unclear which groups will be involved, or -if this can not yet be defined- what the process of selecting these	Their involvement will be through steps 2-4 and we have now added further detail on the sampling in explaining the four steps (pp. 6-7) and also improved text in the Patient and Public Involvement on p.11.

Comment	Actions taken and response ¹
groups is, and what the sampling and recruitment process will be.	
Lastly: the authors are requested to edit the manuscript and address some grammar errors and long sentences to further improve the reading	As recommended, the manuscript has now been edited throughout, and have also simplified the title of the review.